# Early Changes in DCE-MRI Biomarkers May Predict Survival Outcomes in Patients with Advanced Hepatocellular Carcinoma after Sorafenib Failure: Two Prospective Phase II Trials

**DOI:** 10.3390/cancers13194962

**Published:** 2021-10-01

**Authors:** Bang-Bin Chen, Zhong-Zhe Lin, Yu-Yun Shao, Chiun Hsu, Chih-Hung Hsu, Ann-Lii Cheng, Po-Chin Liang, Tiffany Ting-Fang Shih

**Affiliations:** 1Department of Medical Imaging, National Taiwan University Hospital, Taipei City 100, Taiwan; bangbinchen@ntu.edu.tw (B.-B.C.); 006584@ntuh.gov.tw (P.-C.L.); 2Department of Radiology, College of Medicine, National Taiwan University, Taipei City 100, Taiwan; 3Department of Oncology, National Taiwan University Hospital, Taipei City 100, Taiwan; zzlin7460@ntu.edu.tw (Z.-Z.L.); yuyunshao@ntu.edu.tw (Y.-Y.S.); chsu1967@ntu.edu.tw (C.H.); chihhunghsu@ntu.edu.tw (C.-H.H.); alcheng@ntu.edu.tw (A.-L.C.); 4Department of Medical Oncology, National Taiwan University Cancer Center, Taipei City 100, Taiwan; 5Graduate Institute of Oncology, College of Medicine, National Taiwan University, Taipei City 100, Taiwan; 6Department of Internal Medicine, National Taiwan University Hospital, Taipei City 100, Taiwan; 7Department of Medical Imaging, National Taiwan University Hospital Hsin-Chu Branch, Hsin-Chu City 300, Taiwan

**Keywords:** magnetic resonance imaging, hepatocellular carcinoma, survival, axitinib, lenalidomide

## Abstract

**Simple Summary:**

In patients with advanced hepatocellular carcinoma, systemic therapy is recommended by most treatment guidelines. Sorafenib and lenvatinib are both 1st-line treatments for inoperable advanced HCC. Regorafenib, cabozantinib, and ramucirumab have been approved as 2nd-line targeted therapy in patients who show progression or do not tolerate sorafenib. However, there is a lack of imaging biomarkers for predicting survival outcomes in patients receiving 2nd-line targeted therapy after sorafenib failure. In this paper, we try to predict survival outcomes via early changes in the DCE-MRI biomarkers in participants with advanced HCC after 2nd-line targeted therapy following sorafenib failure, taking data from two different prospective clinical trials. We found that an early reduction in tumor perfusion detected by DCE-MRI biomarkers, especially on day 14, may predict survival outcomes in these participants. For the further clinical development of anti-angiogenic therapies, optimal participant selection with predictive biomarkers, such as DCE-MRI, is essential in order to improve treatment outcomes.

**Abstract:**

In this paper, our main objective was to predict survival outcomes using DCE-MRI biomarkers in patients with advanced hepatocellular carcinoma (HCC) after progression from 1st-line sorafenib treatment in two prospective phase II trials. This study included 74 participants (men/women = 64/10, mean age 60 ± 11.8 years) with advanced HCC who received 2nd-line targeted therapy (*n* = 41 with lenalidomide in one clinical trial; *n* = 33 with axitinib in another clinical trial) after sorafenib failure from two prospective phase II studies. Among them, all patients underwent DCE-MRI at baseline, and on days 3 and 14 of treatment. The relative changes (Δ) in the DCE-MRI parameters, including ΔPeak, ΔAUC, and ΔK^trans^, were derived from the largest hepatic tumor. The treatment response was evaluated using the Response Evaluation Criteria in Solid Tumors (RECIST 1.1). The Cox model was used to investigate the associations of the clinical variables and DCE-MRI biomarkers with progression-free survival (PFS) and overall survival (OS). The objective response rate (ORR) was 10.8% (8/74) and the disease control rate (DCR) was 58.1% (43/74). The median PFS and OS values were 1.9 and 7.8 months, respectively. On day 3 (D3), participants with high reductions in ΔPeak_D3 (hazard ratio (HR) 0.4, 95% confidence interval (CI) 0.17–0.93, *p* = 0.017) or ΔAUC_D3 (HR 0.51, 95% CI 0.25–1.04, *p* = 0.043) were associated with better PFS. On day 14, participants with high reductions in ΔPeak_D14 (HR 0.51, 95% CI 0.26–1.01, *p* = 0.032), ΔAUC_D14 (HR 0.54, 95% CI 0.33–0.9, *p* = 0.009), or ΔK^trans^_D14 (HR 0.26, 95% CI 0.12–0.56, *p* < 0.001) had a higher PFS than those with lower reduction values. In addition, high reductions in ΔAUC_D14 (HR 0.53, 95% CI 0.32–0.9, *p* = 0.016) or ΔK^trans^_D14 (HR 0.47, 95% CI 0.23–0.98, *p* = 0.038) were associated with a better OS. Among the clinical variables, ORR was associated with both PFS (*p* = 0.001) and OS (*p* = 0.005). DCR was associated with PFS (*p* = 0.002), but not OS (*p* = 0.089). Cox multivariable analysis revealed that ΔK^trans^_D14 (*p* = 0.002) remained an independent predictor of PFS after controlling for ORR and DCR. An early reduction in tumor perfusion detected by DCE-MRI biomarkers, especially on day 14, may predict favorable survival outcomes in participants with HCC receiving 2nd-line targeted therapy after sorafenib failure.

## 1. Introduction

Hepatocellular carcinoma (HCC) is the sixth most common malignancy and the fourth leading cause of cancer-related death worldwide [1]. In advanced-stage patients, such as those with vascular invasion or distant metastasis, systemic therapy is recommended by most treatment guidelines [2]. Currently, various vascular-endothelial growth factor (VEGF)-targeted anti-angiogenic drugs are available. Sorafenib and lenvatinib are both 1st-line treatments for inoperable advanced HCC [3,4]. Regorafenib, cabozantinib, and ramucirumab have been approved as 2nd-line treatment alternatives in patients who show progression or do not tolerate sorafenib. In addition to anti-angiogenic agents, the immune-programmed cell death protein-1/programmed cell death protein ligand-1 checkpoint inhibitors have recently shown promising outcomes in phase II trials [5]. Recently, combining immunotherapy (atezolizumab) with anti-angiogenic therapy (bevacizumab) has been approved for patients with unresectable or metastatic HCC who have not received prior systemic treatment [6].

Axitinib, a selective inhibitor of VEGF receptor tyrosine kinases 1–3, was approved as a 2nd-line treatment for advanced renal cell carcinoma [7]. Axitinib has been evaluated as a 2nd-line therapy for patients with advanced HCC [8,9]. Lenalidomide, which has both anti-angiogenic and immunomodulatory effects, has also demonstrated efficacy as a 2nd-line treatment for advanced HCC [10,11].

A major challenge in developing anti-angiogenic therapy is identifying the biomarkers needed for predicting treatment efficacy [12,13]. Dynamic contrast-enhanced magnetic resonance imaging (DCE-MRI) is a sensitive method that can be used for detecting tumor blood flow and vascular permeability changes. This method has been actively investigated in cancer clinical trials to explore its clinical potential for monitoring the efficacy of anti-angiogenic therapies [14,15,16,17]. Previous studies have shown that DCE-MRI biomarkers, such as Peak and K^trans^ (volume transfer constant), appear to be capable of predicting therapeutic efficacy in HCC patients receiving anti-angiogenic targeted therapy [18,19,20]. Therefore, the tumor perfusion changes evaluated by DCE-MRI are potential predictive imaging biomarkers for anti-angiogenic therapy in HCC.

There is a lack of imaging biomarkers for predicting survival outcomes in patients receiving 2nd-line targeted therapy after sorafenib failure. Based on previous studies, we hypothesize that the vascular responses detected by DCE-MRI may help us to identify HCC patients who would benefit from 2nd-line targeted therapy. The purpose of this study was to predict survival outcomes via early changes in the DCE-MRI biomarkers in participants with advanced HCC after 2nd-line targeted therapy following sorafenib failure, taking data from two different prospective clinical trials.

## 2. Materials and Methods

We pooled data from two prospective, open label, phase II clinical trials on the efficacy of 2nd-line targeted therapy after sorafenib failure (study 1 on lenalidomide and study 2 on axitinib, Figure 1). Both studies were single-arm investigator-initiated clinical studies and are registered at ClinicalTrials.gov (NCT01273662 and NCT01545804). This study was approved by the Institutional Research Ethics Committee of the National Taiwan University Hospital. Written informed consent was obtained from all participants.

The primary results of these two trials have been previously published [11,21]. Both trials required progression during sorafenib treatment, had consistent eligibility criteria, and collected the same baseline demographic variables. Disease assessments were performed after 4 and 8 weeks of treatment and every 8 weeks thereafter. The response was assessed using the Response Evaluation Criteria in Solid Tumors (RECIST) 1.1. The primary endpoint was the disease control rate (DCR), which was defined as a complete or partial response or as stable disease according to RECIST1.1, meaning there was no progression of tumor-related symptoms for at least eight weeks. The secondary endpoints included the objective response rate (ORR) and PFS according to RECIST1.1, OS, and the alpha-fetoprotein (AFP) response. The objective response was assessed in the subset of patients who had measurable disease and was defined as achieving the best response of either a confirmed or unconfirmed partial or complete response. Disease control was assessed in all patients and was defined as the absence of evidence of progression at the first follow-up disease assessment. PFS was defined as the duration from the date of treatment to the date of documented disease progression or death from any cause. OS was defined as the duration from the date of treatment to the date of death from any cause.

### 2.1. Inclusion and Exclusion Criteria

The inclusion criteria for both studies were as follows: receiving a histological or imaging diagnosis of HCC; having documented progression under treatment with, or intolerance to, sorafenib or other systemic therapies; having an Eastern Cooperative Oncology Group score of 0 or 1; being classified into Child–Pugh class A; and having at least one measurable lesion according to Response Evaluation Criteria in Solid Tumors (RECIST) 1.1. The exclusion criteria were more than one line of systemic therapy, elevated liver function, or recent gastrointestinal bleeding. Lenalidomide (25 mg/day) was administrated on days 1–21 and every 4 weeks (study 1), and axitinib 5 mg was given orally twice daily (study 2) until objective disease progression, the development of unacceptable toxicity, or voluntary discontinuation occurred.

### 2.2. MRI Protocol

Participants were considered feasible for evaluation by DCE-MRI if the target lesions were identified in the liver. All participants underwent a DCE-MRI in two different MRI systems—study 1, 3-T Magnetom Verio (Siemens Healthcare, Erlangen, Germany); study 2, 1.5-T system Signa HD (GE Healthcare, Milwaukee, WI)—at baseline, and on days 3 and 14 of treatment. A dose of 0.1 mmol/kg of gadobutrol (Gadovist^®^, Bayer Healthcare, Berlin, Germany) was injected with an automated injector at a rate of 4 mL/sec followed by a 20 mL saline flush. All participants were asked to hold their breath for as long as they could tolerate and then breathe slowly and smoothly during imaging. The initial 10 s of imaging were used as the baseline images. The DCE-MRI parameters were as follows:

Study 1: Oblique coronal three-dimensional T1-weighted volumetric interpolated breath-hold examination sequence. Section thickness/gap, 5 mm/0 mm; repetition time msec/echo time msec, 2.2/2.5; flip angle, 9°; field of view, 400 × 313 mm; matrix size, 420 × 448; temporal resolution, 6.4 sec; total acquisition time, 170 sec; 600 dynamic images from each participant.

Study 2: Oblique coronal two-dimensional T1-weighted fast spoiled gradient-echo sequence. Section thickness/gap, 8 mm/4 mm; repetition time msec/echo time msec, 3.2/1.1; flip angle, 12°; field of view, 380 × 380 mm; matrix size, 256 × 256; temporal resolution, 3.52 sec; total acquisition time, 110 sec; 720 dynamic images from each participant.

### 2.3. Image Analysis

The DCE-MRI data were analyzed using a commercial software tool (MIStars; Apollo Medical Imaging, Melbourne, Australia). The motion correction algorithm used a 2D rigid body with three adjustable parameters: translation in the x- and y-direction, and in-plane rotation. The semiquantitative parameters Peak (maximal signal intensity minus baseline signal intensity/baseline signal intensity) and AUC (initial area under the gadolinium concentration–time curve at 60 s after contrast injection) were obtained by analyzing the characteristics of the tumor enhancement curves. Furthermore, pharmacokinetic modeling was conducted using a single-input two-compartment model using the aorta as the arterial input function [19,20]. The quantitative parameter K^trans^ (forward volume transfer constant) was automatically calculated pixel by pixel using a constrained non-linear least-squares fitting algorithm with an adjustable delay time. For the tumor perfusion measurement, a region of interest (ROI) was drawn on the perfusion map on a single slice of the largest tumor area, and the follow-up DCE-MRI was selected to be on the same level as the tumor (Figure 2). The hypo-enhanced or necrotic area within the tumor was included. In patients with multiple HCCs, the largest hepatic tumor was chosen for analysis. All ROIs were drawn by the same radiologist. The mean ROI in the tumors was 81.7 ± 51.2 cm^2^.

To evaluate the interobserver variability of these parameters, ROI placement was performed on 40 randomly selected tumors by another radiologist with 27 years of experience in MR imaging.

### 2.4. Statistical Analyses

Data are expressed as means ± standard deviations or as the median with interquartile range when the data were not normally distributed. Interobserver variability was calculated using the intraclass correlation coefficient. The relative changes in Peak, AUC, and K^trans^ were expressed as ΔPeak, ΔAUC, and ΔK^trans^, respectively. The comparison of changes in DCE-MRI biomarkers among subjects on days 3 and 14 according to ORR or DCR was performed with the Mann–Whitney *U* test. The Kaplan–Meier method was used to plot survival curves, and the two-sided log-rank test was used to assess the differences in PFS and OS between the subgroups. Each DCE-MRI biomarker was dichotomized based on the cutoff value determined using the maximally selected rank statistics from the ‘maxstat’ [22] package in the R statistical software (R, version 4.0.2; R Foundation for Statistical Computing, Vienna, Austria). Multivariable analyses were performed using a stepwise forward Cox proportional hazard model that included age, sex, tumor size, AFP, ECOG, liver cirrhosis, vascular invasion, extrahepatic spread, ORR, and DCR. Hazard ratios and 95% confidence intervals (CI) were calculated. The R statistical software and IBM SPSS Statistics software (version 24; IBM Corp., Armonk, NY, USA) were used for the statistical analyses. A *p*-value < 0.05 was considered to indicate a statistically significant difference.

## 3. Results

### 3.1. Participants Characteristics

From April 2011 to March 2016, 74 participants (64 men, 10 women, mean age 60 ± 11.8 years) with advanced HCC were enrolled for DCE-MRI examination. Among them, 41 participants (36 men, 5 women, mean age 59.7 ± 11.6 years) received lenalidomide, while 33 participants (28 men, 5 women, mean age 58 ± 9.7 years) received axitinib treatment (Table 1). All participants had documented disease progression after sorafenib treatment prior to enrollment. Some participants had received prior loco-regional therapy, including surgery (48.6%), ablative therapy (25.7%), or transarterial chemoembolization (81.1%). Using the RECIST 1.1 criteria, the best responses were partial response (PR) in 8 (10.8%) participants, stable disease (SD) in 35 (47.3%) participants, and progressive disease (PD) in 31 (41.9%) participants. The disease control rate and objective response rate were 58.1% (43/74) and 10.8% (8/74), respectively. All participants experienced disease progression and died before April 2018. The median PFS and OS values were 6.9 months (95% CI 2.6–4.7) and 8.6 months (95% CI 8.2–13.2), respectively.

### 3.2. Intraclass Correlation Coefficients of MR Quantitative Parameters

The intraclass correlation coefficients for inter-observer variability were 1.00 (95% CI: 1.00, 0.999) for Peak, 0.999 (95% CI: 0.998, 0.999) for AUC, and 0.998 (95% CI: 0.996, 0.999) for K^trans^.

### 3.3. Comparison of Changes in DCE-MRI Biomarkers between Treatment Groups

All 74 participants underwent a baseline and day 3 (D3) DCE-MRI. Two patients did not receive the DCE-MRI on day 14 (D14) because of their poor general condition.

The changes in the DCE-MRI biomarkers between day 3 and day 14 are shown in Table 2. On day 3, ΔAUC_D3 (*p* = 0.008) and ΔK^trans^_D3 (*p* = 0.024) in the axitinib group were significantly lower than those in the lenalidomide group. On day 14, ΔPeak_D14 (*p* = 0.04), ΔAUC_D14 (*p* = 0.002), and ΔK^trans^_D14 (*p* = 0.003) in the axitinib group were all significantly lower than those in the lenalidomide group (Table 2).

### 3.4. Comparison of Changes in the Imaging Characteristics and DCE-MRI Biomarkers According to Treatment Response

There were no significant differences in imaging characteristics (tumor size, cirrhosis, vascular invasion, or extrahepatic spread) when the participants were grouped according to their treatment response (Appendix A).

There were no significant differences in the changes in the DCE-MRI biomarkers when participants were grouped according to ORR (all *p* > 0.05). However, ΔK^trans^_D14 was significantly decreased in participants with PR or SD compared to participants with PD (*p* = 0.01). The area under the receiver operating characteristic was 0.68, with a sensitivity of 0.63 and a specificity of 0.76 when using −11.8% as the cutoff value for ΔK^trans^_D14 (Appendix A).

### 3.5. Factors Associated with PFS and OS

Among clinical variables, ORR was associated with both PFS (*p* = 0.001) and OS (*p* = 0.005). DCR was associated with PFS (*p* = 0.002), but not OS (*p* = 0.089) (Table 3).

On day 3, participants with high reductions in ΔPeak_D3 (HR 0.4, 95% CI 0.17–0.93, *p* = 0.017) or ΔAUC_D3 (HR 0.51, 95% CI 0.25–1.04, *p* = 0.043) were associated with better PFS. On day 14, participants with high reductions in ΔPeak_D14 (HR 0.51, 95% CI 0.26–1.01, *p* = 0.032), ΔAUC_D14 (HR 0.54, 95% CI 0.33–0.9, *p* = 0.009), or ΔK^trans^_D14 (HR 0.26, 95% CI 0.12–0.56, *p* < 0.001) had a longer PFS than those with low reduction values (Figure 3). In addition, high reductions in ΔAUC_D14 (HR 0.53, 95% CI 0.32–0.9, *p* = 0.016) or ΔK^trans^_D14 (HR 0.47, 95% CI 0.23–0.98, *p* = 0.038) were associated with a longer OS (Figure 4). Cox multivariable analysis revealed that ΔK^trans^_D14 (HR 0.29, 95% CI 0.14–0.63, *p* = 0.002) remained an independent predictor of PFS after controlling for ORR and DCR.

## 4. Discussion

This study, based on two different prospective clinical trials, found that an early reduction in DCE-MRI biomarkers may predict survival outcomes in participants with HCC using a 2nd-line targeted therapy after sorafenib failure. On day 3, high reductions in ΔPeak_D3 or ΔAUC_D3 were associated with longer PFS. On day 14, high reductions in ΔPeak_D14, ΔAUC_D14, or ΔK^trans^_D14 were associated with a longer PFS. In addition, high reductions in ΔAUC_D14 or ΔK^trans^_D14 were associated with a longer OS. When ORR and DCR were incorporated in the multivariable analysis, ΔK^trans^_D14 (*p* = 0.002) remained an independent predictor of PFS. Therefore, DCE-MRI can be used to identify possible responders to targeted therapy as early as on day 3 and to predict PFS as early as 14 days post-treatment in these patients. These results were achievable regardless of the MR vendor and magnetic strength (1.5 vs. 3.0 Tesla).

We found a higher reduction in DCE-MRI biomarkers in the axitinib group than in the lenalidomide group on both day 3 and day 14. These results may be explained by the different pharmacokinetic pathways of these two drugs. Axitinib is a pure anti-angiogenesis drug, whereas lenalidomide has both anti-angiogenic and immunomodulatory effects. It appears that the anti-angiogenesis effect of axitinib was stronger than that of lenalidomide, which could be reflected by the high early reduction in DCE-MRI biomarkers. However, in a 2nd-line therapy setting, the high anti-angiogenesis effect of Axitinib may not necessarily lead to higher tumor shrinkage or longer survival outcomes than with lenalidomide because all tumors already showed progression after the 1st-line angiogenic agent (sorafenib). Therefore, the drug parameter was not significant in the univariate analysis of PFS or OS. Currently, whether DCE-MRI could detect perfusion changes by immunomodulatory drugs or immunotherapy is still unknown. Nevertheless, we could anticipate survival benefits in both groups if a high reduction in DCE-MRI biomarkers was achieved after treatment.

We found no significant differences between DCE-MRI changes and RECIST response (ORR). These results could imply that these two evaluation methods represent different pathophysiological aspects of the tumors in response to targeted therapy. The RECIST response only reflected changes in tumor size without information on vascularity or tumor necrosis. In contrast, DCE-MRI biomarkers reflect the global perfusion changes in the tumors [19,23], and their correlation with tissue markers of tumor hypoxia in participants with primary liver cancer has been previously reported [24]. Our study suggests that tumor perfusion reduction precedes tumor size shrinkage in the responders to targeted therapy. The potential correlation between DCE-MRI biomarkers and other response evaluation methods (such as modified RECIST) is worthy of further investigation.

We found that the DCE-MRI biomarkers may predict survival outcomes in patients receiving the 2nd-line targeted therapy, consistent with previous studies for HCC patients receiving the 1st-line systemic treatment [19,20]. For example, a retrospective study of 92 participants with advanced HCC has found that participants with a high ΔPeak reduction within one week following systemic treatment had a longer OS (*p* = 0.023) compared with participants with a low Peak reduction [20]. Another study has shown that the percentage of K^trans^ change after treatment with sorafenib and metronomic tegafur/uracil is an independent predictor of tumor response, PFS, and OS [18]. Thus, in respondent participants, tumor perfusion reduction induced by anti-angiogenic agents could be detected as early as 3 days after treatment, and post-treatment day 14 would be an appropriate timepoint to predict therapeutic efficacy in these participants.

Several issues may limit the clinical applicability of DCE-MRI for predicting the therapeutic efficacy of anti-angiogenic therapy in HCC. First, contrast enhancement patterns are usually heterogeneous in tumors, and the investigators’ selection of the ROI may introduce bias into the measurements [25]. An independent imaging review may be needed to validate the results demonstrated in the lenvatinib trial [26]. More objective or computer-aided methods of target lesion selection should be developed to improve the reproducibility of measurements. Second, the biological mechanism accounting for the changes in each imaging biomarker after anti-angiogenic therapy remains elusive. Preclinical models may help reveal the pharmacodynamic correlation of these imaging biomarkers in the tumor microenvironment, explore the optimal time points of imaging evaluation, and compare the predictive values for treatment efficacy [27,28].

Our study has several limitations. First, our sample size was small due to the phase II clinical trial study design. Additional studies with larger groups of participants are necessary in order to validate our findings. Second, we did not include other DCE-MRI parameters because only Peak and K^trans^ have been previously reported to correlate with OS [18,20,29]. Third, although lenalidomide and axitinib are both classified as a targeted therapy, they have different molecular pathways and treatment efficacies in advanced HCC. We included treatment groups as univariable and multivariable variables to eliminate bias due to treatments, but we did not discover significant survival differences between the two groups.

In conclusion, we found that an early reduction in tumor perfusion detected by DCE-MRI biomarkers, especially on day 14, may predict survival outcomes in participants with HCC receiving 2nd-line targeted therapy after sorafenib failure. For the further clinical development of anti-angiogenic therapies, optimal participant selection with predictive biomarkers, such as DCE-MRI, is essential in order to improve treatment outcomes.

## Figures and Tables

**Figure 1 cancers-13-04962-f001:**
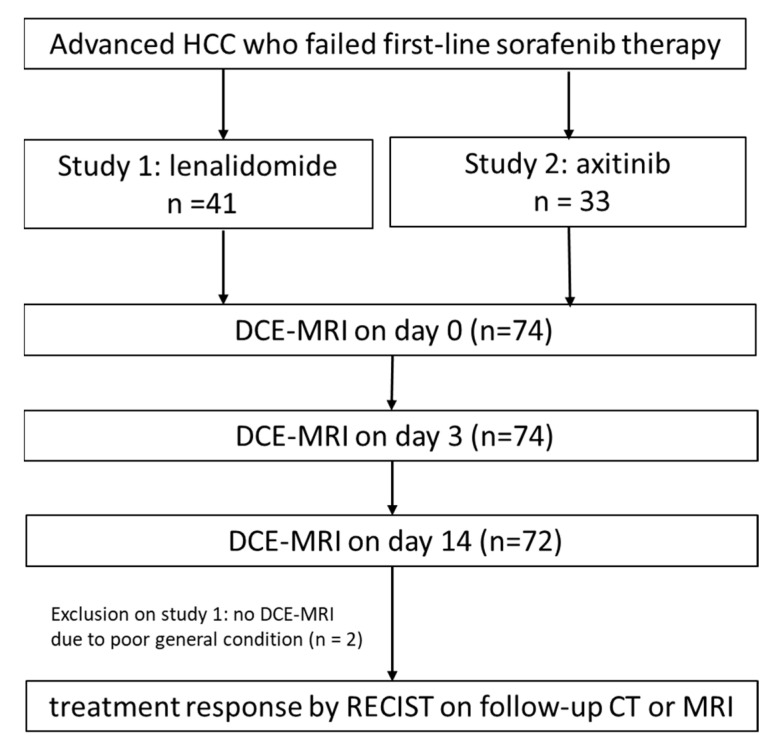
Study flowchart presenting a summary of the final study participants. HCC = hepatocellular carcinoma; RECIST = Response Evaluation Criteria in Solid Tumors; DCE = dynamic contrast enhanced.

**Figure 2 cancers-13-04962-f002:**
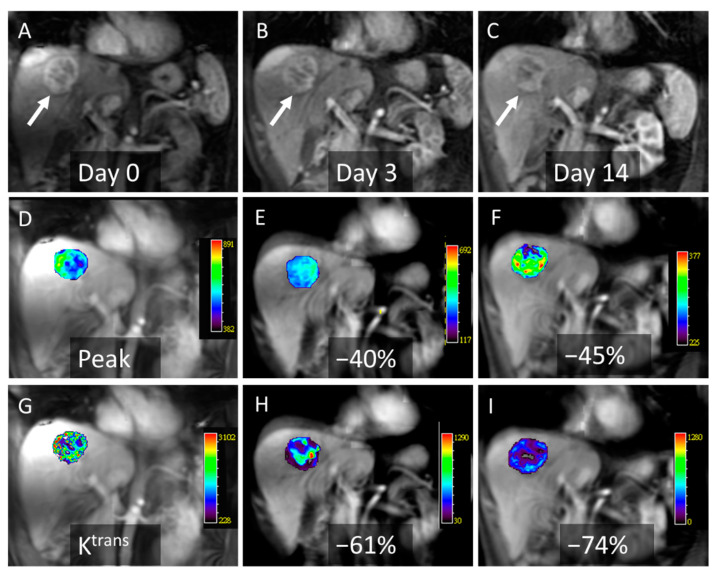
Images from a 51-year-old man who received axitinib with a progression-free survival of 0.9 months and an overall survival of 13 months. (**A**) A coronal contrast-enhanced T1-weighted image depicting a hepatocellular carcinoma in the right upper lobe (arrow) on day 0 (baseline, **A**), day 3 (**B**), and day 14 (**C**). Serial changes in Peak compared to the baseline value (**D**) show that there is a marked decrease on day 3 (−40%, (**E**)) and day 14 (−45%, (**F**)). Serial changes in K^trans^ also show a marked decrease on day 3 (−61%, (**E**)) and day 14 (−74%, (**F**)) when compared to the baseline (**G**).

**Figure 3 cancers-13-04962-f003:**
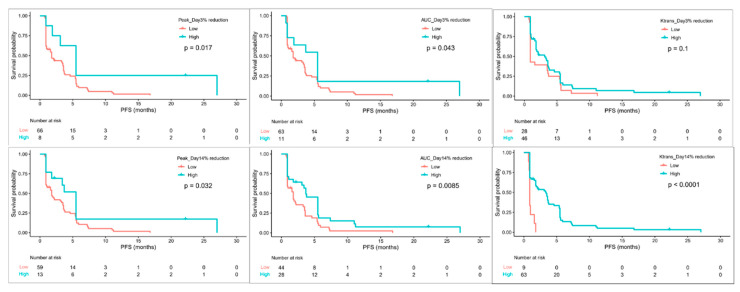
Kaplan–Meier survival curves of PFS grouped by changes in DCE-MRI biomarkers on day 3 and day 14.

**Figure 4 cancers-13-04962-f004:**
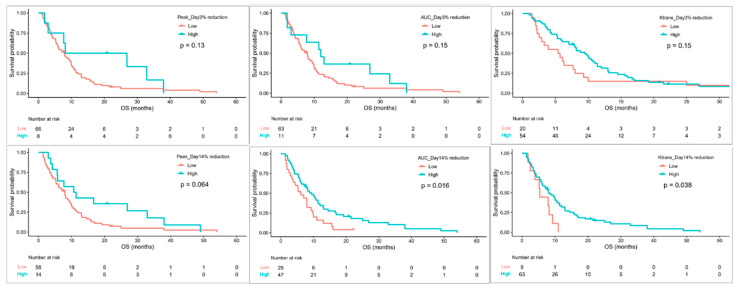
Kaplan–Meier survival curves of OS grouped by changes in DCE-MRI biomarkers on day 3 and day 14.

**Table 1 cancers-13-04962-t001:** Patient characteristics and treatment response.

Characteristics	Lenalidomide	Axitinib	*p*-Value	All Participants
Number	41	33		74
Age, year (mean, SD)	59.7 (11.6)	58 (9.7)	0.825	60 (11.8)
Sex (women)	5 (12.2)	5 (15.2)	0.978	10 (13.5)
Tumor size (mean, SD)	78.6 (51.3)	85.6 (51.6)	0.564	81.7 (51.2)
Etiology				
Hepatitis B	26 (63.4)	25 (75.8)	0.375	
Hepatitis C	8 (19.5)	6 (18.2)	1	
Alcoholic	4 (9.8)	2 (6.1)	0.88	
ECOG			0.059	
0	7 (17.1)	13 (39)		20 (27.0)
1	34 (82.9)	20 (60.6)		54 (73.0)
AFP > 400 ng/mL	27 (65.9)	14 (42.4)	0.059	41 (55.4)
Liver cirrhosis	31(75)	24 (73)	0.172	55(74.3)
Vascular invasion	22 (53.7)	19 (57.6)	0.919	41 (55.4)
Extrahepatic spread	35 (85.4)	28 (84.8)	1	63 (85.1)
Child–Pugh = 5			<0.001 *	
5	23 (43.7)	33 (100)		55 (74.3)
6	19 (46.3)	0		19 (25.7)
BCLC stage			0.195	
B	0	2 (6)		
C	41 (100)	31 (94)		
Prior therapy				
Surgery	19 (46.3)	17 (51.5)	0.835	36 (48.6)
Ablation	8 (19.5)	11 (33.3)	0.278	19 (25.7)
TACE	33 (80.5)	27 (81.8)	1	60 (81.1)
RECIST 1.1 response			0.105	
PR	6 (14.6)	2 (6.1)		8 (10.8)
SD	15 (36.6)	20 (60.6)		35 (47.3)
PD	20 (48.8)	11 (33.3)		31 (41.9)

Note: Unless otherwise indicated, data are reported as numbers, and data in parenthesis are percentages. ECOG = Eastern Cooperative Oncology Group; AFP = alpha-fetoprotein; BCLC = Barcelona Clinic Liver Cancer; TACE = transarterial chemoembolization; RECIST = Response Evaluation Criteria in Solid Tumors; PR = partial response; SD = stable disease; PD = progressive disease. *: *p*-value indicates a significant difference.

**Table 2 cancers-13-04962-t002:** Comparison of changes in the DCE-MRI parameters between lenalidomide and axitinib between day 3 and day 14.

DCE-MRI Parameters	Lenalidomide	Axitinib	*p*-Value	All
Number of participants on Day 3	41	33		74
ΔPeak_D3 (%)	−3.7 (−11.9, 5.2)	−7.5 (−17.8, 3.6)	0.265	−4.3 (−13, 3.8)
ΔAUC_D3 (%)	−8 (−18.9, 23.7)	−25.7 (−53.4, 7.3)	0.008 *	−12 (−37.4, 1)
ΔK^trans^_D3 (%)	−0.9 (−40.8, 74.1)	−30.6 (−64.2, −8.8)	0.024 *	−21.4 (−49.6, 28.5)
Number of participants on Day 14	39	33		72
ΔPeak_D14 (%)	−2.9 (−11.7, 5.7)	−11.9 (−22.4, 1.4)	0.04 *	−4 (−15.9, 4)
ΔAUC_D14 (%)	−3.1 (−24.8, 20.2)	−39.1 (−52.25, −5.9)	0.002 *	−19.5 (−43.8, 11.5)
ΔK^trans^_D14 (%)	−7.7 (−45.3, 45.8)	−47.8 (−71.6, −14.2)	0.003 *	−22.3 (−57.6, 26.3)

Note: The data are expressed as the median and interquartile range. * *p*-value indicates a significant difference.

**Table 3 cancers-13-04962-t003:** Factors associated with PFS and OS based on univariable and multivariable analyses.

Survival outcomes	PFS				OS			
	Univariate		Multivariate		Univariate		Multivariate	
Parameter	HR (95 % CI)	*p*-Value	HR (95 % CI)	*p*-Value	HR (95 % CI)	*p*-Value	HR (95 % CI)	*p*-Value
Drug (axitinib vs. lenalidomide)	0.79 (0.49–1.28)	0.347			0.84 (0.52–1.37)	0.488		
Age (>60 vs. ≤60 y/o)	1.03(0.64–1.67)	0.89			0.97 (0.64–1.58)	0.915		
Sex (man vs. woman)	1.62 (0.82–3.2)	0.166			1.19 (0.58–2.44)	0.636		
Size	1.15 (0.72–1.85)	0.559			1.17 (0.71–1.93)	0.528		
AFP (>400 ng/mL)	0.99 (0.62–1.59)	0.956			1.37 (0.85–2.23)	0.198		
ECOG	0.87 (0.51–1.49)	0.622			0.89 (0.51–1.56)	0.687		
Liver cirrhosis	0.82 (0.62–1.08)	0.151			0.84 (0.63–1.12)	0.241		
Vascular invasion	0.93 (0.58–1.5)	0.778			1.12 (0.70–1.80)	0.638		
Extrahepatic spread	1.21 (0.63–2.32)	0.566			1.23 (0.60–2.50)	0.569		
ORR	0.19 (0.07–0.49)	0.001 *	0.2 (0.07–0.52)	0.001 *	0.26 (0.1–0.66)	0.005 *	0.29 (0.11–0.76)	0.012 *
DCR	0.51 (0.31–0.84)	0.002 *			0.65 (0.39–1.01)	0.089		
DEC-MRI parameters								
ΔPeak_D3 (%)	0.4 (0.17–0.93)	0.017 *			0.55 (0.25–1.21)	0.13		
ΔAUC_D3 (%)	0.51 (0.25–1.04)	0.043 *			0.61 (0.31–1.21)	0.15		
ΔK^trans^_D3 (%)	0.67 (0.44–1.13)	0.1			0.68 (0.4–1.15)	0.15		
ΔPeak_D14 (%)	0.51 (0.26–1.01)	0.032 *			0.6 (0.3–1.04)	0.064		
ΔAUC_D14 (%)	0.54 (0.33–0.9)	0.009 *			0.53 (0.32–0.9)	0.016 *	0.63 (0.37–1.07)	0.085
ΔK^trans^_D14 (%)	0.26 (0.12–0.56)	<0.001 *	0.29 (0.14–0.63)	0.002 *	0.47 (0.23–0.98)	0.038 *		

Note: ECOG = Eastern Cooperative Oncology Group; HR = hazard ratio; CI = confidence interval; ORR = objective response rate; DCR = disease control rate. * *p*-value indicates a significant difference.

## Data Availability

The data presented in this study are available on request from the corresponding author.

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
