# Peer review of "Early Changes in DCE-MRI Biomarkers May Predict Survival Outcomes in Patients with Advanced Hepatocellular Carcinoma after Sorafenib Failure: Two Prospective Phase II Trials"

_cancers, 2021, doi:10.3390/cancers13194962_

Round 1

Reviewer 1 Report

Authors intended that early tumor shrinkage after 2nd line chemotherapy induction could be an excellent prognostic marker of advanced hepatocellular carcinomas.

The major problem of the study is the combination of two different chemotherapy studies. Both reagents work as anti-vascularization drugs. However, most of the DCE-MRI parameters differed between Lenalidomide-treated cases and Axitinib-treated cases (Table 2). Basically, Axitinib-treated cases showed better early tumor shrinkage. Then, I cannot understand why the drug parameter did not show a significant factor in univariate analysis (Table 3). Would you please clarify it? It seems better to compare DCE-MRI parameters in each treatment group.

Please explain how to deal with multiple HCC cases.

The cut-off of the AFP marker is relatively high (>400 ng/ml) in Table 3. Please clarify how to define the cut-off value. Also, the PIVKA-II tumor marker can be applied to this univariate and multivariate analysis because it is more associated with tumor size or viability.

Reviewer 2 Report

The paper is well writen and presented and there is scientific data presented are sound. Many more studies are required however before this method is incorporated in clinical practice. 

Author Response

Thank you for the positive comments.

Reviewer 3 Report

General

Authors investigate various MRI biomarkers obtained from contrast enhanced scans in the early course of anti-vascular therapy. The major methodological concern derives from pooling the results of the two trials, obtained on different MRI machines with differing field strengths and MRI protocols. I suggest to publish the imaging results for the two trials separately from each other, which will make the results much more clear.

2nd drawback: As the authors state correctly DCE MRI has been successfully applied for response assessment of targeted therapies for HCC. Therefore, the additional amount of knowledge obtained from this study is small.

Abstract

Obviously the authors use data from two prospective trials. it remains however unclear, which was the major content of the two trials. Were axitinib and lenalidomide used in both of the trials or one medication in either one of the studies. This already should become clear within the abstract.

In the conclusion the authors claim DCE may predict ‘better’ survival outcomes…better as compared to what? Please clarify

Methods

2.1 ‘histologically or clinically diagnosis of HCC’…what does clinical mean? By Imaging?

Imaging data are summarized from two considerably different MRI machines, 1.5t and 3T. These data cannot be either pooled or compared. If pooling is done then appropriate analysis has to be provided to proof that results/data of both systems are similar. As such, without any validation of the different methodologies, pooling is not appropriate. Also, MRI protocols differed substantially from each other, especially different total acquisition times.

Essentially the authors here report results of different studies in one manuscript. As stated above results should be published separately from each other.

Please view an cite: 34467453

Measurement of only the biggest lesion may not provide a comprehensive picture, why not evaluate at least three lesions per patient?

HCC measured by RECIST 1.1. is not the adequate readout, please provide analysis according to mRECIST.

Round 2

Reviewer 1 Report

All my concerns were appropriately answered. 

Author Response

Thank you for the positive comments.

Reviewer 3 Report

The atuhor's answers are not suffcient.

Essentially none of the requests for additional analyses (like adding mRECIST to RECIST 1.1) has been followed and the major methodological flaws mentioned in the comments remain!

Author Response

Response to Reviewer 3 Comments (Round 2)

Dear editor and reviewer:

Point 1: The author's answers are not sufficient. Essentially none of the requests for additional analyses (like adding mRECIST to RECIST 1.1) has been followed, and the major methodological flaws mentioned in the comments remain!

Response 1:

We sincerely appreciate the valuable comments from the reviewer. We have explained the concerns raised by the reviewer and made revisions according to the suggestions in the previous letter (Round 1).

For example, we explained several reasons and advantages for combining these two investigator-initiated prospective clinical trials.

Because RECIST 1.1. was chosen in our clinical trial protocols as endpoint evaluation, we used RECIST 1.1 response in the univariate and multivariate analysis. A recent study on Atezolizumab plus Bevacizumab for advanced HCC (N Engl J Med. 2020) also used RECIST 1.1 response. This issue is also mentioned in our manuscript: “The potential correlation between DCE-MRI biomarkers and other response evaluation methods (such as modified RECIST) is worthy of further investigation.” We sincerely thank the reviewer’s suggestion and will use both methods in our future clinical trials.

To our knowledge, whether DCE-MRI could be helpful in a 2nd-line therapy setting is still unknown. Current 2nd-line agents after sorafenib failure are limited in HCC. The knowledge obtained from this study could help drug development, patient selection, and early outcomes prediction in participants with HCC receiving 2nd-line targeted therapy in the future. We hope our revised manuscript could fulfill the high academic requirements of your journal. If you have any questions, please do not hesitate to contact us. Thank you very much for your consideration.

Sincerely,

Tiffany Ting-Fang Shih

Department of Medical Imaging

National Taiwan University Hospital